# Hierarchical Position Embedding of Graphs with Landmarks and Clustering for Link Prediction

## ABSTRACT

Learning positional information of nodes in a graph is important for link prediction tasks. We propose a representation of positional information using representative nodes called landmarks. A small number of nodes with high degree centrality are selected as landmarks, which serve as reference points for the nodes' positions. We justify this selection strategy for well-known random graph models, and derive closed-form bounds on the average path lengths involving landmarks. In a model for scale-free networks, we prove that landmarks provide asymptotically exact information on inter-node distances. We apply theoretical insights to practical networks, and propose Hierarchical Position embedding with Landmarks and Clustering (HPLC). HPLC combines landmark selection and graph clustering, where the graph is partitioned into densely connected clusters in which nodes with the highest degree are selected as landmarks. HPLC leverages the positional information of nodes based on landmarks at various levels of hierarchy such as nodes' distances to landmarks, inter-landmark distances and hierarchical grouping of clusters. Experiments show that HPLC achieves state-of-the-art performances of link prediction on various datasets in terms of HIT@K, MRR, and AUC.

## CCS CONCEPTS

• **Computing methodologies** → **Learning latent representations**.

## KEYWORDS

Link Prediction, Network Science, Graph Neural Networks

ACM Reference Format:
Anonymous Author(s). 2024. Hierarchical Position Embedding of Graphs with Landmarks and Clustering for Link Prediction. In *Proceedings of Under Review (WWW '24)*. ACM, New York, NY, USA, 15 pages. https://doi.org/XXXXXXX.XXXXXXX

## 1 INTRODUCTION

Graph Neural Networks are foundational methods for various graph related tasks such as node classification [18, 23, 27, 44], link prediction [2, 26, 50], graph classification [47], and graph clustering [35]. In this paper, we focus on the task of link prediction using GNNs.

*WWW '24, May 13–17, 2024, SINGAPORE*
© 2024 Association for Computing Machinery.
ACM ISBN 978-1-4503-XXXX-X/18/06...$15.00
https://doi.org/XXXXXXX.XXXXXXX

Message passing GNNs (MPGNNs) [18, 23, 27, 44] have been successful in learning structural node representations through neighborhood aggregation. However, MPGNNs achieved subpar performances on link prediction tasks due to their inability to distinguish isomorphic nodes. Subgraph-based methods [34, 50] were proposed for link prediction based on dynamic node embeddings from enclosing subgraphs to tackle isomorphism. Labeling methods [31, 51] assigns node labels to identify each node via hashing or aggregation functions. Meanwhile, position-based methods [15, 30, 45, 49] have been proposed to represent the positions of nodes in a graph, which enables differentiating isomorphic nodes according to their positions and enhances the expressive power of GNNs [41, 49].

The position of a node can be defined using its distances to other nodes. Position-based methods are beneficial to link prediction tasks thanks to several properties e.g., the connectivity of a node pair may be closely related to their relative distances, and isomorphic nodes can be differentiated by their positions. Bourgain [9] showed that inter-node distance information can be encoded into low-dimensional embeddings. Linial [32] constructed Bourgain's embeddings based on nodes' distances to random node sets to obtain $O(\log^2 N)$-dimensional embeddings. P-GNN [49] realized Linial's method by computing and aggregating messages from random sets. However, its performance falls short of state-of-the-art and does not scale well [45]. Another line of positional encoding used the eigenvectors of graph Laplacian [14, 30]. However, Laplacian methods have stability issues [45] and may not outperform methods based on structural features [50]. Thus, there is significant room for improving both performance and scalability of positional node embedding for link prediction.

In this paper, we propose an effective and efficient representation of the nodes' positional information. We select a small number of representative nodes called *landmarks* and impose hierarchy on the graph by associating each node with a landmark. Each node computes distances to the landmarks, and each landmark computes distances to other landmarks. Such distance information is combined so as to represent nodes' positional information. Importantly, we select landmarks and organize the graph in a principled way, unlike previous methods using random selection, e.g., [9, 32, 49].

The key question is how to select landmarks. Our choice is to select nodes with *high degree* where the degree of a node is often used as a measure of its importance or centrality. The choice is motivated from the theory of *network science*. In network models with preferential attachment (PA) [5], nodes with high degrees, called *hubs*, play a central role in characterizing the network. PA is a process such that, if a new node joins the graph, it is more likely to connect to nodes with higher degrees. Node degrees follow power-law distribution, and the network exhibits scale-free property [5]. Hubs are abundant in social/citation networks and World Wide Web. Real-world networks may not be exactly scale-free; however, the analysis of network models gives insights into design of algorithms

that work quite well in practice. A similar approach, i.e., the analysis of tractable models applied to design of practical algorithms, is taken for shortest path problems in road networks [1].

We provide a theoretical justification of landmark selection based on degree centrality for a well-known class of random graphs [5, 16]. We show that the inter-node distances are well-represented by the *detour* via landmarks. In networks with preferential attachment, we show that the strategy of choosing high-degree nodes as landmarks is *asymptotically optimal* in the following sense. The minimum distance among the detours via landmarks is asymptotically equal to the shortest path distance. We show even a small number of landmarks relative to network size, e.g., $O(\log N)$, suffice to achieve optimality. This proves that the hub-type landmarks offer short paths to nodes, manifesting *small-world phenomenon* [5, 46]. In addition, we show that in models where hubs are absent, one can reduce the detour distance by selecting a higher number of landmarks.

Motivated by the theory, we propose **H**ierarchical **P**osition embedding with **L**andmarks and **C**lustering (**HPLC**). HPLC partitions graph into $O(\log N)$ clusters which are locally dense and appoints the node with the highest degree in the cluster as the landmark. Our intention is to bridge gap between theory and practice: hubs may not be present in practical networks. Thus it is important to distribute the landmarks evenly over the network so that nodes can access nearby/local landmarks, instead of merely choosing the highest-degree landmarks. Next, we form a graph of higher hierarchy, i.e., the graph of landmarks, and compute its Laplacian encoding based on the inter-landmark distances. The encoding is assigned as a *membership* to the nodes belonging to the cluster, so as to learn positional features at the cluster level. We further optimize our model using the encoding based on hierarchical grouping of clusters. The computation of HPLC can be mainly done during preprocessing, incurring low computational costs. We perform experiments on 7 datasets, comparing HPLC with 16 baseline methods. We show that HPLC achieves state-of-the-art performances in most cases, demonstrating effectiveness and robustness over prior position-/distance-based methods.

Our contributions are summarized as: 1) we propose HPLC, a novel algorithm for link prediction using hierarchical position embedding based on landmarks combined with graph clustering; 2) building upon network science, we derive closed-form bounds on average path lengths via detours for well-known random graphs which, to our belief, are important theoretical findings; 3) we conduct extensive experiments to show that HPLC achieves state-of-the-art performances in most cases.

## 2 RANDOM GRAPHS WITH LANDMARKS

### 2.1 Notation

We consider undirected graph $G = (V, E)$ where $V$ and $E \subseteq V \times V$ denote the set of vertices and edges, respectively. Let $N$ denote the number of nodes in the graph, or $N = |V|$. $\mathbf{A} \in \mathbb{R}^{N \times N}$ denotes the adjacency matrix of $G$. $d(v, u)$ denotes the geodesic (shortest-path) distance between $v$ and $u$. Node attributes are defined as $X = \{x_1, ..., x_N\}$ where $x_i \in \mathbb{R}^n$ denotes the feature vector of node $i$. We consider methods of embedding $X$ into latent space $Z = \{z_1, ..., z_N\}, z_i \in \mathbb{R}^m$. We study the node-pair-level task of predicting the link probability between node embedding $z_u$ and $z_v$.

### 2.2 Representation of Positions using Landmarks

The distances between nodes provide rich information on the graph structures. For example, a connected and undirected graph can be represented by a finite metric space with its vertex set and the inter-node geodesic distances. However, computing and storing shortest paths for all pairs of nodes incur high complexity. We select a small number of representative nodes called *landmarks*, and hierarchically organize the graph based on the nodes' distances to landmarks. The landmark selection and hierarchical organization are done in a principled way, which is in contrast to previous methods [9, 32, 49] which utilizes distances to *random* subset of nodes.

The landmarks are denoted by $\lambda_1, \cdots, \lambda_K \in V$ where $K$ denotes the number of landmarks. For node $v$, we define $K$-dimensional vector of distances to landmarks:

$$D(v) := (d(v, \lambda_1), d(v, \lambda_2), \cdots, d(v, \lambda_K))$$

An overview of our method is as follows. Each node is assigned to a landmark. To represent the position of node $v$ assigned to landmark $\lambda$, we will use the vector of distances $D(v)$ as well as the position information of $\lambda$ relative to other landmarks. We explore various levels of hierarchy induced by landmarks in the graph, which we explain in detail in Sec. 3. The central element in our approach is the vector of distances to landmarks, $D(\cdot)$.

The key question is, how much information $D(\cdot)$ has on the inter-node distances within the graph. From triangle inequality, the distance between nodes $u$ and $v$ are bounded as

$$d(u, v) \leq \min_{i=1,...,K} \left[ d(u, \lambda_i) + d(\lambda_i, v) \right]$$

which states that the *detour* via landmarks, i.e., from $u$ to $\lambda_i$ to $v$, is longer than $d(u, v)$, but the shortest detour, or the minimum component of $D(u) + D(v)$, may provide a good estimate of $d(u, v)$. The key design questions are: how to select *good* landmarks, and how many of them? Drawing upon the theory of *network science*, we analyze a well-known class of random graphs, derive the average path lengths associated with landmarks, and glean design insights from the analysis.

### 2.3 Path lengths via Landmarks in random networks

The framework by [17] provides a useful tool for analyzing path lengths for a wide range of classes of random networks. Following the framework, the probability of the existence of an edge for node $i$ and $j$ denoted by $q_{ij}$ is given by

$$q_{ij} = \frac{h_i h_j}{\beta} \tag{3}$$

where $\beta$ is a parameter depending on the network model. $h_i$ is *tag* information of node $i$, and is related to the connectivity or degree of the node.

From the continuum approximation [4], tag information $h$ is regarded as a continuous random variable (RV) with distribution $\rho(\cdot)$. For some function $f$, $\langle f(h) \rangle$ denotes the expectation of $f(h)$ of a node chosen at random. $\langle f(h) \rangle_Q$ denotes the expectation of $f(h)$ of *landmarks* chosen under some distribution $Q$.

$$P(L_{ij} > s) = \exp\left[-\frac{h_i h_j}{\beta N \langle h^2 \rangle} \cdot \langle h^2 \rangle_Q K(N) \cdot (s-1)\left(\frac{\langle h^2 \rangle N}{\beta}\right)^{s-1}\right], \quad s = 1, 2, \cdots \tag{1}$$

$$\bar{l} \le \frac{-2\langle \log h \rangle - \log\left(\langle h^2 \rangle_Q K(N)\right) + \log(N\beta\langle h^2 \rangle) + \log\log\left(\frac{N\langle h^2 \rangle}{\beta}\right) - \gamma}{\log N + \log\langle h^2 \rangle - \log\beta} + \frac{1}{2} \tag{2}$$

THEOREM 1. *Let $L_{ij}$ denote the random variable representing the minimum path length from node $i$ to $j$ among the detours via $K(N)$ landmarks. The landmarks are chosen i.i.d. according to distribution $Q$. Asymptotically in $N$, $P(L_{ij} > s)$ is given by* (1).

The proof of Theorem 1 is in Appendix A.1. In (1), the design parameters are $\langle h^2 \rangle_Q$ and $K(N)$ which are related to what kind of landmarks are chosen, and how many of them, respectively. Next, we bound the average of the minimum path length among the detours via landmarks.

THEOREM 2. *Assume $K(N) = o(N)$ and $K(N) \to \infty$ as $N \to \infty$. The mean of the minimum path length among the detours via landmarks, denoted by $\bar{l}$, is bounded above as* (2) *where $\gamma \approx 0.5772$ is the Euler's constant.*

The proof of Theorem 2 is in Appendix A.2. The assumption $K(N) = o(N)$ implies that the number of landmarks is chosen to be not too large compared to $N$. We apply the results to some well-known random graph models.

### 2.4 Erdős-Rényi Model

The Erdős-Rényi (ER) model [16] is a classical random graph in which every node pair is connected with a common probability. For ER graphs, model parameters $\beta$ and $h$ can be set as $\beta = \langle k \rangle N$ and $h \equiv \langle k \rangle$ respectively, where $\langle k \rangle$ denotes the mean degree of nodes [8]. From (3), we have $q_{ij} = \langle k \rangle / N$, i.e., the probability of edge formation between any node pair is constant. Thus, the node degree follows the Poisson distribution with mean $\langle k \rangle$ for large $N$. Assume $\langle k \rangle$ is a finite constant. From (2), The mean of the minimum of path length via landmarks in ER network, denoted by $\bar{l}_{\text{ER}}$, is bounded above as

$$\bar{l}_{\text{ER}} \le \frac{2\log N - \log K(N)}{\log\langle k \rangle} \tag{6}$$

asymptotically in $N$. The average length of the shortest paths in ER graphs without landmarks, denoted by $\bar{l}_{\text{ER}}^*$, is given by [8, 17]

$$\bar{l}_{\text{ER}}^* = \frac{\log N}{\log\langle k \rangle} \tag{7}$$

By comparing (6) and (7), we observe that the detour via landmarks incurs the overhead of at most factor 2. This is because nodes in ER graphs appear *homogeneous*, and thus the path length to and from landmarks are on average similar to the inter-node distance. Thus the average distance of a detour will be twice the direct distance. However, (6) implies that the *minimum* detour distance can be reduced by using multiple ($K(N) > 1$) landmarks. The reduction

can be substantial, e.g., if $K(N) = N^{1-\varepsilon}$ for some $\varepsilon \in (0, 1)$, we have, from (6),

$$\bar{l}_{\text{ER}} \le (1 + \varepsilon) \cdot \bar{l}_{\text{ER}}^*$$

For example, selecting $\sqrt{N}$ landmarks guarantees a 1.5-factor approximation of the shortest path distance. By making $\varepsilon$ close to 0, we get arbitrarily close to the shortest path distance.

**Discussion.** Due to having Poisson distribution, the degrees in ER graphs are highly concentrated on mean $\langle k \rangle$. There seldom are nodes with very large degrees, i.e., most nodes look alike. Thus, the design question should be on *how many* rather than on *what kind* of landmarks. We benefit from choosing a large number of landmarks, e.g., $K(N) = N^{1-\varepsilon}$. However, there is a trade-off: the computational overhead of managing $K(N)$-dimensional vector $D(v)$ will be high.

### 2.5 Barabási-Albert Model

The Barabási-Albert (BA) model [5] generates random graphs with preferential attachment. BA networks are characterized by continuous growth over time: initially there are $m$ nodes, and new nodes arrive to the network over time. Due to preferential attachment, the probability of the connection of the existing node to the newly arriving node is *proportional* to its degree. The probability of an edge in BA networks is shown to be [8]

$$q_{ij} = \frac{m}{2}\frac{1}{\sqrt{t_i t_j}}$$

with $h_i = 1/\sqrt{t_i}$ and $\beta = \frac{m}{2}$, where $t_i$ is the time of arrival of node $i$. Since the probability of a newly arriving node connecting to node $i$ is proportional to $h_i$, the degree of nodes with large $h_i$ is likely to be high. For large $N$, the distribution of $h$ is derived as [17]

$$\rho(h) = \frac{2}{N}h^{-3}, \ h \in \left[\frac{1}{\sqrt{N}}, 1\right]. \tag{8}$$

By applying $\rho(\cdot)$ to (2), we bound the average path length with landmarks denoted by $\bar{l}_{\text{BA}}$ as (4).

**Selecting Landmarks with Large $h$ is Optimal**. Unlike ER graphs, there exists a landmark selection strategy which achieves the asymptotically optimal distance, despite using a small number of landmarks relative to the network size. Specifically, selecting landmarks from a pool of nodes with large $h$ values is optimal.

THEOREM 3. *Suppose $K(N)$ landmarks are randomly selected from top-$(\log N) \cdot K(N)$ nodes with the largest values of $h$, assuming $K(N)$ satisfies the condition in Theorem 2. Assuming $m = O(1)$, the average path length via landmarks, or $\bar{l}_{\text{BA}}$, is bounded above by* (5) *asymptotically in $N$.*

$$\bar{l}_{\text{BA}} \le \frac{\log N - \log(\langle h^2 \rangle_Q K(N)) + \log\log N + \log\log\log N + \log\left[2\log(m/2)/m\right]}{\log\log N + \log(m/2)} + \frac{1}{2}. \tag{4}$$

$$\bar{l}_{\text{BA}} \le \frac{\log N - \log\log K(N) + 2\log\log N}{\log\log N} \tag{5}$$

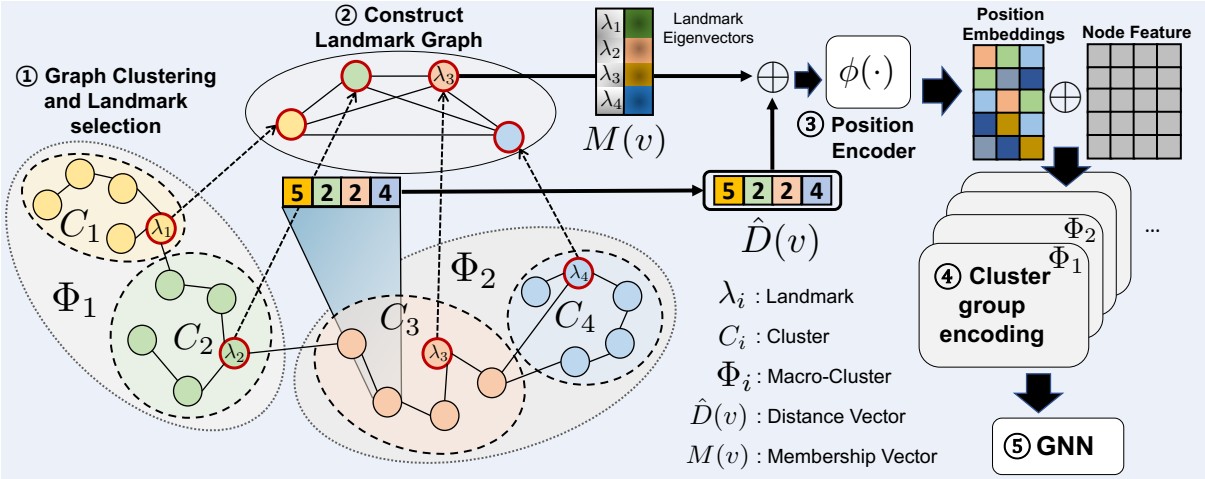

**Figure 1: Overview of HPLC.** ① **Partition the graph into** $K$ **clusters using FluidC, select landmarks based on degrees, and compute distance vectors of nodes.** ② **Construct a landmark graph to compute membership vectors based on eigenvectors of graph Laplacian.** ③ **Compute positional embeddings by combining membership and distance vectors and passing them through an encoder.** ④ **Concatenate positional embeddings and node features, and project them onto cluster-group embedding spaces.** ⑤ **Neighborhood aggregation using GNNs.** ⊕ **denotes concatenation.**

The proof of Theorem 3 is in Appendix A.3. We claim the optimality of the strategy in Theorem 3 in the following sense. Since Theorem 2 states that $K(N)$ is slowly increasing in $N$, e.g., $K(N) = O(\log N)$, a feasible strategy for Theorem 3 is to choose $O(\log N)$ landmarks from top−$O(\log^2 N)$ degree nodes. For such $K(N)$, the numerator of the RHS of (5) is $\approx \log N$ for large $N$. Meanwhile, the average length of shortest paths in BA networks is given by [12]

$$\bar{l}_{\text{BA}}^* = \frac{\log N}{\log \log N}$$

From (5), we conclude that $\bar{l}_{BA}$ *is asymptotically equal to* $\bar{l}_{BA}^*$. This implies that the shortest detour via landmarks under our strategy has the same length as the shortest path on average in the asymptotic sense.

**Discussion.** The node degree in BA networks is known to follow *power-law distribution*, which predicts the emergence of hubs. Inter-node distances can be drastically reduced due to the presence of hubs, known as *small-world phenomenon* [5]. Our analysis shows that the shortest path between two nodes are indeed well-approximated by the detour via landmarks chosen from high-degree nodes. Importantly, this is achieved even without large number of landmarks, say we let $K(N) = O(\log N)$ in (5).

**Summary of Analysis.** ER and BA models represent two contrasting cases of degree distributions. The degree distribution of ER graph is highly concentrated, i.e., the variation of node degrees is small, or hubs are absent. By contrast, the degree of BA graphs follows power-law distribution, i.e., the variation of node degree is large, and hubs are present. Our analysis shows that $(1 + \epsilon)$-factor approximation (ER) and asymptotic optimality (BA) are achievable by detour via landmarks. The derived bounds are numerically verified via simulation in Appendix B.

## 2.6 Design Insights from Theory

The key design parameters in our analysis are $\langle h^2 \rangle_Q$ and $K(N)$ in (2).

**Landmark Selection and Graph Clustering.** In order to make $\langle h^2 \rangle_Q$ large, one may choose landmarks with sufficiently large $h$, e.g., high-degree nodes. In practice, however, such high-degree nodes may not always provide short paths, unlike scale-free networks. For example, suppose all the hubs are located at one end of the network. The nodes at the other end of the network have to make a long detour via landmarks, even in order to reach nodes in local neighborhoods.

Thus, in order to better capture local graph structures, we propose to partition the graph into *clusters* such that the nodes within a cluster tend to be densely connected, i.e., close to one another. Then we pick the node with the highest degree within each cluster as the landmark, as suggested by the analysis. Each landmark represents the associated cluster, and the distance between nodes can be captured by distance information associated with the cluster landmarks. We empirically find that such combination of clustering and landmark selection yields improved results.

**Number of Landmarks.** Although large number of landmarks $K(N)$ appears preferable, our analysis show that $K(N)$ does *not* drastically reduce distances, unless $K(N)$ is very large, e.g., $N^{1-\epsilon}$. A large number of landmarks can hamper scalability. Thus, we use only a moderate number of landmarks, and empirically find that setting $K(N) = O(\log N)$ suffices to yield good results.

## 3 PROPOSED METHOD

In this section, **H**ierarchical **P**osition embedding with **L**andmarks and **C**lustering (**HPLC**) is described. The main method combining landmarks and clustering is explained in Sec. 3.1. Additional

optimization methods leveraging landmarks and clusters are introduced, which are membership encoding (Sec. 3.2) and cluster-group encoding (Sec. 3.3). An overview of HPLC is depicted in Fig. 1.

## 3.1 Graph Clustering and Landmark Selection

*Graph clustering* is a task of partitioning the vertex set into disjoint subsets called clusters, where the nodes within a cluster tend to be densely connected relative to those between clusters [40]. We will use FluidC graph clustering algorithm proposed in [35] as follows. Suppose we want to partition $G$ to $K$ clusters. FluidC initially selects $K$ random central nodes, assigns each node to clusters $C_1, \cdots, C_K$, and iteratively assigns nodes to clusters according to the following rule. Node $u$ is assigned to cluster $P_{k^*(u)}$ such that

$$k^*(u) = \underset{k=1,\ldots,K}{\operatorname{argmax}} \frac{|\{u, \mathcal{N}(u)\} \cap C_k|}{|C_k|} \quad (9)$$

where $\mathcal{N}(u)$ denotes the neighbors of $u$. This assignment rule prefers community candidates of small sizes (denominator), which results in well-balanced cluster sizes. The rule also prefers communities already containing many neighbors of $u$ (numerator), which makes clusters locally dense. As a result, FluidC generates densely-connected cohesive clusters of relatively even sizes.

In addition, FluidC has complexity $O(|E|)$, and thus is highly scalable [35]. Importantly, the number of clusters $K$ can be specified beforehand, which is crucial because $K$ is a hyperparameter in our method. By contrast, in widely-used Louvain's algorithm [7], it is difficult to control $K$, and a comparison is provided in Appendix F.

For each cluster, we select the node with the highest degree as the landmark, where $\lambda_k$ denotes the landmark of cluster $C_k$. For each node, we compute the distances to landmarks $\lambda_1, \cdots, \lambda_K$ to yield $K$-dimensional vector of distances. Since $G$ may not be a connected graph, we define the following:

$$\hat{d}(v, \lambda_k) = \begin{cases} d(v, \lambda_k), & \text{if a path exists from } v \text{ to } \lambda_k, \\ d_{\max} + 1, & \text{otherwise.} \end{cases} \quad (10)$$

where $d_{\max}$ is defined as follows. For each node $v$, we compute the distances to the landmarks within the connected component containing $v$ and set $d_{\max}$ to the maximum among all the computed distances. The *distance vector* (DV) of node $v$, denoted by $\hat{D}(v)$, is given by $\hat{D}(v) = (\hat{d}(v, \lambda_1), \hat{d}(v, \lambda_2), \ldots, \hat{d}(v, \lambda_K))$. As mentioned earlier, we set number of clusters $K = \eta \log N$ where integer $\eta$ is a hyperparameter. Experiments on the values of $\eta$ is in Appendix D.

**Effect of Clustering on Landmarks.** The proposed method first performs clustering, and then selects landmarks based on degree centrality. The analysis in Sec. 2 showed that the nodes with sufficiently high degree centrality should be chosen as landmarks. The question is whether the landmarks chosen after clustering will have sufficiently high degrees.

A supporting argument for incorporating clustering into analysis can be made for BA networks. Theorem 3 states that, choosing $K(N) = O(\log N)$ landmarks from top-$(\log N) \cdot K(N) = O(\log^2 N)$ nodes in degrees is optimal. We show that, $\log N$ landmarks chosen after FluidC clustering are indeed within top-$(\log N)^2$ degree centrality through simulation. Table 1 compares the rank of degrees of landmarks selected through FluidC clustering versus the rank of top-$(\log N)^2$ degree nodes in BA networks. We observe that the

| Network size $N$ | rank: cluster landmarks | rank: top-$(\log N)^2$ | Fraction of landmarks within rank top-$(\log N)^2$ |
|---|---|---|---|
| 500 | 5.02% | 6.38% | 100% |
| 1000 | 3.02% | 4.37% | 100% |
| 2000 | 1.76% | 2.85% | 100% |
| 5000 | 0.90% | 1.48% | 100% |

Table 1: Rank of degrees of landmark nodes in BA networks. For example, if $N = 500$, all the landmarks selected from clustering are within the top 5.02%-degree nodes.

landmarks selected after clustering have sufficiently large degrees for optimality, i.e., all of their degrees are within top-$(\log N)^2$.

The result can be explained as follows. Since the number of clusters is relatively small ($O(\log N)$) in our method, each cluster contains a relatively large number of nodes. Scale-free networks are known to have *modules* which are densely connected subgraphs and are sparsely connected to each other [19]. A proper clustering algorithm is expected to detect modules. Due to scale-free property, such modules are likely to be connected to larger modules. Thus, each cluster is likely to contain nodes with sufficiently high degrees which are, according to Theorem 3, good candidates for landmarks.

In practical networks, one can expect similar effects from FluidC clustering. Given the small number of clusters as input, the algorithm will detect locally dense clusters resulting in landmarks which have high degrees and are evenly distributed over the network. Thus, landmark selection after clustering can be a good heuristic motivated by the theory.

## 3.2 Membership Encoding with Graph Laplacian

For the nodes within the same cluster, we augment the nodes' embeddings with information identifying that they are members to the same community, which we call *membership*. The membership is extracted from landmarks, and is based on the relative positional information among landmarks. Thus, not only the nodes within the same cluster have the same membership, but also the nodes of neighboring clusters have "similar" membership. The membership of a node is encoded into a *membership vector* (MV) which is combined with DV in computing the node embeddings.

We consider the graph consisting only of landmark nodes, and use graph Laplacian [6] to encode their relative positions. A complete graph of landmarks is constructed where the edge weight between landmarks $u$ and $v$ is set to $e_{uv} = \exp(-\hat{d}(u, v)^2/T)$ where $T$ denotes normalizing parameter of heat kernel. Let $\hat{A} \in \mathbb{R}^{K \times K}$ denote the weighted adjacency matrix. The normalized graph Laplacian is given by $L = I - \Delta^{-\frac{1}{2}} \hat{A} \Delta^{-\frac{1}{2}}$ where degree matrix $\Delta$ is given by $\Delta_{ii} = \sum_j \hat{A}_{ij}$. The eigenvectors of $L$ are used as MVs, i.e., for node $v \in C_k$, the MV of node $v$, denoted by $M(v)$, is the eigenvector of $L$ associated with landmark $\lambda_k$.

MV provides positional information in addition to DV, using the graph of upper hierarchy, i.e., landmarks. We use a random flipping of the signs of eigenvectors to resolve ambiguity [15]. A problem with Laplacian encoding is the time complexity, of which the eigen-decomposition is cubic in network size. Previous approaches used a subset of eigenspectrum [15, 30]. However, the landmark graph has

small size, i.e., $K = O(\log N)$. Thus, our model can exploit the *full* spectrum, enabling accurate representation of landmark graphs.

## 3.3 Cluster-group Encoding

We propose *cluster-group encoding* as an additional model optimization. Several neighboring clusters are further grouped into a *macro-cluster*, i.e., a cluster of clusters. The embeddings of nodes in a macro-cluster use an encoder specific to that macro-cluster. The motivation is that, using a separate encoder per local region may facilitate capturing attributes specific to local structures or latent features of the community, which is important for link prediction.

For cluster $C_k$, let $\Phi_{i(k)}$ denote the macro-cluster which contains $C_k$ where $i(k)$ denote the index of the macro-cluster. The node embedding $z_v$ for node $v$ is computed as

$$z_v = f_{i(k)}\left(x_v \oplus \phi\left(M(v) \oplus \hat{D}(v)\right)\right), \quad v \in C_k \subset \Phi_{i(k)},$$

where $x_v$ is the input node feature, $M(v)$ and $\hat{D}(v)$ are MV and DV, $\phi$ is membership/distance encoder, and $\oplus$ denotes concatenation. $f_{i(\cdot)}$ denotes the encoder specific to macro-cluster $\Phi_{i(\cdot)}$. We set the number of macro-cluster denoted by $R$ such that $K$ is a multiple of $R$, and each macro-cluster contains $K/R$ clusters. Cluster-group encoding requires $R$ encoders, one per macro-cluster. To limit the model size, we empirically set $R = \min(15, \lfloor K/\eta \rfloor)$.

Finally, output embedding $z_v$ is input to GNN layers given by

$$h_v^l = \text{GNN}(h_v^{l-1}, \mathbf{A}), \quad \text{for } l = 1, 2, \cdots$$

where $h_v^0 = z_v$. HPLC can be combined with different types of GNNs, and the related study is provided in Sec. 7.2.

## 4 PROPERTY OF HPLC AS NODE EMBEDDING

In [41], the *(positional) node embeddings* are formally defined as:

*Definition 4.* The node embeddings of a graph with adjacency matrix $\mathbf{A}$ and input attributes $X$ are defined as joint samples of random variables $(Z_i)_{i \in V} | \mathbf{A}, X \sim p(\cdot | \mathbf{A}, X), Z_i \in \mathbb{R}^d, d \geq 1$, where $p(\cdot | \mathbf{A}, X)$ is a $\mathcal{G}$-equivariant probability distribution on $\mathbf{A}$ and $X$, that is, $\pi(p(\cdot | \mathbf{A}, X)) = p(\cdot | \pi(\mathbf{A}), \pi(X))$ for any permutation $\pi(\cdot)$.

It is argued in [41] that positional node embeddings are good at link prediction tasks. The postional embeddings preserve the relative positions of nodes, thus can differentiate isomorphic nodes according to their positions and identify the closeness between nodes. We claim that HPLC qualifies as positional node embeddings according to Definition 4 as follows.

In HPLC, landmark-based distance function, eigenvectors of graph Laplacian with random flipping, and graph clustering method are $\mathcal{G}$-equivariant functions of $\mathbf{A}$ ignoring the node features $X$. The eigenvectors of graph Laplacian are permutationally equivariant under the node permutation, i.e., switching of corresponding rows/-columns of adjacency matrix. Also, the Laplacian eigenmap can be regarded as a $\mathcal{G}$-equivariant function in the sense of expectation, if it is combined with random sign flipping of eigenvectors [41]. In case of multiplicity of eigenvalues, we can slightly perturb the edge weights of landmark graphs to obtain simple eigenvalues [36]. The graph clustering in HPLC is a randomized method, because initially $K$ central nodes are selected at random. Thus, embedding output $Z$ is a function of $\mathbf{A}, X$, and random noise, which proves our claim.

Also, HPLC has higher expressive power than traditional message-passing GNNs. HPLC is trained to predict the link between a node pair based on their positional embeddings. The embeddings are learned over the joint distribution of distances from node pairs to common landmarks which are spread out globally over the graph. By contrast, traditional GNNs learn the embeddings based on the marginal distributions of local neighborhoods of node pairs. By a similar argument to Sec. 5.2 of [49] based on mutual information, we conclude that HPLC is more expressive than traditional GNNs.

## 5 COMPLEXITY ANALYSIS

### 5.1 Time Complexity

We first consider the time complexity of computing $\hat{D}(v)$ for all $v \in V$. There are $\eta \log N$ landmarks, and for each landmark, computing the distances from all $v \in V$ to the landmark requires $O(|E| + N \log N)$ using Fibonacci heap. Thus, the overall complexity is $O(|E| \log N + N(\log N)^2)$. Next, we compute Laplacian eigenvectors of landmark graph, where its time complexity is $O(K^3) = O((\log N)^3)$. The computation of $\hat{D}(v)$ and $M(v)$ are done once, and thus can be considered as a preprocessing step. Overall, the time complexity of HPLC is low, and our experiments shows that HPLC handles large or dense graphs well.

### 5.2 Space Complexity

The space complexity of HPLC is mainly from the GNN models for computing the node embeddings. Additional space complexity of HPLC is from the membership/distance encoder $\phi(\cdot)$ which is $O((F + R)H_{\text{in}})$, and cluster-group encoding which is $O(H_{\text{in}}H_{\text{out}}R)$ respectively, where $F$ denotes the node feature dimension, $R = \min(15, \lfloor K/\eta \rfloor)$ denotes the number of macro-clusters, $H_{\text{in}}$ denotes the hidden dimension of $\phi(\cdot)$, and $H_{\text{out}}$ denotes the hidden dimension of cluster-group encoders.

Overall, the time and space complexity of HPLC is quite reasonable. We show this by comparing the actual resource usage between vanilla GCN and HPLC in Appendix C.

## 6 EXPERIMENTS

### 6.1 Experimental setting

**Datasets.** Experiments were conducted on 7 datasets widely used for evaluating link prediction. For experiments on small graphs, we used PubMed, Cora, Citeseer, and Facebook. For experiments on dense or large graphs, we chose DDI, COLLAB, and CITATION2 provided by OGB [24]. Detailed statistics and evaluation metrics associated with the datasets are provided in Table 10 in Appendix G.
**Baseline models.** We compared HPLC with Adamic Adar (AA) [2], Matrix Factorization (MF) [29], Node2Vec [21], GCN [27], Graph-SAGE [23], GAT [44], P-GNN [49], NBF-net [54], and plug-in type approaches like JKNet [48], SEAL [50], GCN+DE [31], GCN+LPE [15], GCN+LRGA [38], Graph Transformer+LPE [14], and PEG-DW+ [45]. All methods except for AA and GAE are computed by the same decoder, which is a 2-layer MLP. For a fair comparison, we use GCN in most plug-in type approaches: SEAL, GCN+DE, GAE, JKNet, GCN+LRGA, GCN+LPE, and HPLC.
**Evaluation metrics.** Link prediction was evaluated based on the ranking performance of positive edges in the test data over negative

| Baselines | Avg. (H.M.) | CITATION2 | COLLAB | DDI | PubMed | Cora | Citeseer | Facebook |
|---|---|---|---|---|---|---|---|---|
| Adamic Adar | 65.74 (50.65) | 76.12 ± 0.00 | 53.00 ± 0.00 | 18.61 ± 0.00 | 66.89 ± 0.00 | 77.22 ± 0.00 | 68.94 ± 0.00 | 99.41 ± 0.00 |
| MF | 54.06 (42.65) | 53.08 ± 4.19 | 38.74 ± 0.30 | 17.92 ± 3.57 | 58.18 ± 0.01 | 51.14 ± 0.01 | 50.54 ± 0.01 | 98.80 ± 0.00 |
| Node2Vec | 64.01 (51.17) | 53.47 ± 0.12 | 41.36 ± 0.69 | 21.95 ± 1.58 | 80.32 ± 0.29 | 84.49 ± 0.49 | 80.00 ± 0.68 | 86.49 ± 4.32 |
| GCN (GAE) | 76.35 (66.67) | 84.74 ± 0.21 | 44.14 ± 1.45 | 37.07 ± 5.07 | 95.80 ± 0.13 | 88.68 ± 0.40 | 85.35 ± 0.60 | 98.66 ± 0.04 |
| GCN (MLP) | 76.48 (67.53) | 84.79 ± 0.24 | 44.29 ± 1.88 | 39.31 ± 4.87 | 95.83 ± 0.80 | 90.25 ± 0.53 | 81.47 ± 1.40 | 99.43 ± 0.02 |
| GraphSAGE | *78.51 (71.48)* | 82.64 ± 0.01 | 48.62 ± 0.87 | 44.82 ± 7.32 | 96.58 ± 0.11 | 90.24 ± 0.34 | 87.37 ± 1.39 | 99.29 ± 0.01 |
| GAT | - | - | 44.14 ± 5.95 | 29.53 ± 5.58 | 85.55 ± 0.23 | 82.59 ± 0.14 | 87.29 ± 0.11 | 99.37 ± 0.00 |
| JKNet | - | - | 48.84 ± 0.83 | 57.98 ± 7.68 | 96.58 ± 0.23 | 89.05 ± 0.67 | 88.58 ± 1.78 | 99.43 ± 0.02 |
| P-GNN | - | - | - | 1.14 ± 0.25 | 87.22 ± 0.51 | 85.92 ± 0.33 | 90.25 ± 0.42 | 93.13 ± 0.21 |
| GTrans+LPE | - | - | 11.19 ± 0.42 | 9.22 ± 0.20 | 81.15 ± 0.12 | 79.31 ± 0.09 | 77.49 ± 0.02 | 99.27 ± 0.00 |
| GCN+LPE | 74.60 (67.00) | 84.85 ± 0.35 | 49.75 ± 1.35 | 38.18 ± 7.62 | 95.50 ± 0.13 | 76.46 ± 0.15 | 78.29 ± 0.21 | 99.17 ± 0.00 |
| GCN+DE | 72.48 (60.04) | 60.30 ± 0.61 | 53.44 ± 0.29 | 26.63 ± 6.82 | 95.42 ± 0.08 | 89.51 ± 0.12 | 86.49 ± 0.11 | 99.38 ± 0.02 |
| GCN+LRGA | 78.42 (74.47) | 65.05 ± 0.22 | 52.21 ± 0.72 | *62.30 ± 9.12* | 93.53 ± 0.25 | 88.83 ± 0.01 | 87.59 ± 0.03 | 99.42 ± 0.05 |
| SEAL | 77.08 (62.88) | 85.26 ± 0.98 | *53.72 ± 0.95* | 26.25 ± 8.00 | 95.86 ± 0.28 | 92.55 ± 0.50 | 85.82 ± 0.44 | *99.60 ± 0.02* |
| NBF-net | - | - | - | 4.03 ± 1.32 | *97.30 ± 0.45* | *94.12 ± 0.17* | 92.30 ± 0.23 | 99.42 ± 0.04 |
| PEG-DW+ | 81.67 (75.42) | *86.03 ± 0.53* | 53.70 ± 1.18 | 47.88 ± 4.56 | 97.21 ± 0.18 | 93.12 ± 0.12 | *94.18 ± 0.18* | 99.57 ± 0.05 |
| **HPLC** | **85.77 (82.39)** | **86.15 ± 0.48** | **56.04 ± 0.28** | **70.03 ± 7.02** | **97.38 ± 0.34** | **94.95 ± 0.18** | **96.15 ± 0.19** | **99.69 ± 0.00** |

**Table 2: Link prediction results on various datasets. All baselines and our method were evaluated for 10 repetitions. Bold denotes the best performance, and *Italic* indicates the second best performance. We used a single NVIDIA RTX 3090 with 24GB memory on all datasets except CITATION2 and A100 GPU with 40GB memory on CITATION2. - indicates 'out-of-memory' (OOM). Some baselines suffered from OOM on large graphs due to the high memory usage from storing a large number of shortest paths, attention weights, or aggregation of hidden embedding vectors, etc. Similar OOM results as well as poor performance of those baselines were reported in [45] and [52]. For SEAL and GCN+DE, we trained 2% of training data and evaluated 1% of both validation and test set respectively on CITATION2. We trained 15% of training data but evaluated all of the validation and test sets on COLLAB. Both implementations followed the guideline on the official GitHub of SEAL-OGB. 'Avg.' denotes the average of performance metrics, and 'H.M' indicates their harmonic mean. (-) means that we do not report the average and harmonic mean due to OOM.**

ones. For COLLAB and DDI, we ranked all positive and negative edges in the test data, and computed the ratio of positive edges which are ranked in top-$k$. We did not utilize validation edges for computing node embeddings when we predicted test edges on COLLAB. In CITATION2, we computed all positive and negative edges, and calculated the reverse of the mean rank of positive edges. Due to high complexity when evaluating SEAL, we only trained 2% of training set edges and evaluated 1% of validation and test set edges respectively, as recommended in the official GitHub of SEAL. For Cora, Citeseer, PubMed, and Facebook, we utilized Area Under ROC Curve (AUC). If applicable, we calculated the average and harmonic mean (HM) of the measurements. HM penalizes the model for very low scores, thus is a useful indicator of robustness.

**Hyperparameters.** We used GCN as our base GNN encoder. MLP is used in decoders, except GAE. All baselines do not use edge weights. The details of hyperparameters are provided in Appendix H.

## 6.2 Results

Experimental results are summarized in Table 2. HPLC outperformed the baselines on most datasets. HPLC achieved large performance gains over GAE combined with GCN on all datasets, which are 88.9% on DDI, 27.0% on COLLAB, 12.7% on Citeseer, 7.1% on Cora, and 1.4% on CITATION2. HPLC showed superior performance over SEAL, achieving gains of 167% on DDI, and 12.0% on Citeseer. We compare HPLC with other distance-based methods. Compared to GCN+DE which encodes distances from a target node set whose

representations are to be learned, or to P-GNN which uses distances to random anchor sets, HPLC achieved higher performance gains by a large margin. Compared to other positional encoding methods such as GCN+LPE, Graph Transformer+LPE, and PEG-DW+, HPLC also achieved performance gains from all datasets. The results show that approximate inter-node distances via landmarks can be effective for representing positional information of nodes.

SEAL and NBF-net performed poorly on DDI which is a highly dense graph. Since the nodes of DDI have a large number of neighbors, the enclosing subgraphs are both very dense and large, and the model struggles with learning the representations of local structures or paths between nodes. By contrast, HPLC achieved the best performance on DDI, demonstrating its effectiveness on densely connected graphs.

Finally, we computed the average and harmonic means of performance measurements except for the methods with OOM problems. Although the averages are taken over heterogeneous metrics and thus do not represent specific performance metrics, they are presented for comparison purposes. In summary, HPLC achieved the best average and harmonic mean of performance measurements, demonstrating both its effectiveness and robustness.

## 7 ABLATION STUDY

In this section, we provide ablation study. Additional studies on different types of node centrality and clustering algorithms are relegated to Appendix E and F, respectively, due to limited space.

| DV | CE | MV | COLLAB | DDI | PubMed | Cora | Citeseer |
|----|----|----|--------|-----|--------|------|----------|
| ✔ | ✗ | ✗ | 53.31 ± 0.62 | 46.86 ± 9.91 | 95.37 ± 0.13 | 92.32 ± 0.25 | 95.13 ± 0.18 |
| ✔ | ✔ | ✗ | 55.56 ± 0.37 | 68.75 ± 7.43 | 96.67 ± 0.18 | 93.94 ± 0.21 | 95.83 ± 0.15 |
| ✔ | ✔ | ✔ | **56.04 ± 0.28** | **70.03 ± 7.02** | **97.38 ± 0.34** | **94.95 ± 0.18** | **96.15 ± 0.19** |

**Table 3: Ablation study of each component.**

| Dataset | w/ GraphSAGE | w/ GAT | w/ GCN |
|---------|--------------|--------|--------|
| PubMed | 96.63 (+0.05) | 93.18 (+7.63) | **97.38** (+1.55) |
| Cora | **96.18** (+5.94) | 91.93 (+9.34) | 94.95 (+4.0) |
| Citeseer | 94.97 (+7.60) | 94.60 (+7.31) | **96.15** (+14.68) |
| Facebook | 99.48 (+0.19) | 99.40 (+0.03) | **99.69** (+0.26) |

**Table 4: Ablation study on GNN types for HPLC. + denotes the performance gain over the default encoder for each GNN, i.e., without HPLC.**

## 7.1 Model Components

Table 3 shows the performances in the ablation analysis for the model components. The components denoted by "DV", "CE" and "MV" columns in Table 3 indicate the usage of *distance vector*, *cluster-group encoding* and *membership vector*, respectively. "DV" is the default component of HPLC. We observe that the performance is indeed improved by adding components "CE" and "MV" to HPLC. This shows that all the hierarchical components of HPLC contribute to the performance improvement.

## 7.2 Combination with various GNNs

HPLC can be combined with different GNN encoders. We experimented the combination with three GNN encoders. Table 4 shows that, HPLC enhances the performance of various types of GNNs.

## 8 RELATED WORK

**Link Prediction.** Earlier methods for link prediction used heuristics [2, 53] based on manually designed formulas. GNNs were subsequently applied to the task, e.g., GAE [26] is a graph auto-encoder which reconstructs adjacency matrices combined with GNNs, but cannot distinguish isomorphic nodes. SEAL [50] is proposed as structural link representation by extracting enclosing subgraphs and learning structural patterns of those subgraphs. The authors demonstrated that higher-order heuristics can be approximately represented by lower-order enclosing subgraphs thanks to $\gamma$-decaying heuristic. Multi-scale link learning [10] was proposed to learn enclosing subgraphs at various scales. LGLP [11] used the graph transformation prior to GNN layers for link prediction, and Walk Pooling [34] proposed to learn subgraph structure based on random walks. However, the aforementioned methods need to extract enclosing subgraphs of edges and compute their node embeddings on the fly. CFLP [52] is a counterfactual learning framework for link prediction to learn causal relationships between nodes. However, its time complexity is $O(N^2)$ for finding counterfactual links with nearest neighbors.

**Distance- and Position-based Methods.** P-GNN [49] proposed position-aware GNN based on distances for injecting positional

information into node embeddings. P-GNN focuses on realizing Bourgain's embedding [9] guided by Linial's method [32] and performs message computation and aggregation based on distances to random subset of nodes. By contrast, we judiciously select representative nodes in combination with graph clustering and use the associated distances. Laplacian positional encodings [6, 15] use eigenvectors of graph Laplacian as positional embeddings in which positional features of nearby nodes are encoded to be similar to one another. Graph transformer was combined with positional encoding learned from Laplacian spectrum [14, 30]. However, transformers with full attention have high computational complexity and do not scale well for link predictions in large graphs. Distance encoding (DE) as node labels was proposed and its expressive power was analyzed in [31]. In [51], the authors analyzed the effects of various node labeling tricks using distances. However, these two methods do not utilize distances as positional information.

**Networks with landmarks.** Algorithms augmented with landmarks have actively been explored for large networks, where the focus is mainly on estimating the inter-node distances [13, 33, 42] or computing shortest paths [3, 20, 39, 43]. An approximation theory on inter-node distances using embeddings derived from landmarks is proposed in [28] which, however, based on randomly selected landmarks, whereas we analyze detour distances under a judicious selection strategy. Landmarks were used for efficient heuristics for finding shortest paths, e.g., the ALT algorithm [20] which exploits preprocessed distances to landmarks to derive lower bounds. Notably in [37], vectors of distances to landmarks are used to estimate inter-node distances, and landmark selection strategies based on various node centralities are proposed. However, the work did not provide theoretical analysis on the distances achievable under detours via landmarks. The aforementioned works do not consider landmark algorithms in relation to link prediction tasks.

## 9 CONCLUSION

We proposed a hierarchical positional embedding method using landmarks and graph clustering for link prediction. We provided a theoretical analysis of the average distances of detours via landmarks for well-known random graphs. From the analysis, we gleaned design insights on the type and number of landmarks to be selected and proposed HPLC which effectively infuses positional information using $O(\log N)$-landmarks for the link prediction on real-world graphs. Experiments demonstrated that HPLC achieves state-of-the-art performance and has better scalability as compared to existing methods on various graph datasets of diverse sizes and densities. In the future, we plan to analyze the landmark strategies for various types of random networks and extend HPLC to other graph-related tasks such as graph classification, generation, etc.

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

$$p_{ij}^\lambda(s) = 1 - \exp\left[-\left\{\sum_{v_1=1}^N \cdots \sum_{v_{s-1}=1}^N q_{iv_1}q_{v_1v_2}\cdots q_{v_{s-1}j} - \sum_{\substack{v_1=1 \\ v_1\neq\lambda}}^N \cdots \sum_{\substack{v_{s-1}=1 \\ v_{s-1}\neq\lambda}}^N q_{iv_1}q_{v_1v_2}\cdots q_{v_{s-1}j}\right\}\right] \quad (11)$$

## A  PROOFS

### A.1  Proof of Theorem 1.

The framework from [17] is used for the proof, and their technique is briefly described as follows. We first state a key lemma from [17]:

LEMMA 1. *If $A_1, \cdots, A_n$ are mutually independent events and their probabilities fulfill relations $\forall_i P(A_i) \leq \varepsilon$,*

$$P\left(\bigcup_{k=1}^n A_k\right) = 1 - \exp\left(-\sum_{k=1}^n P(A_k)\right) - R$$

*where $0 \leq R < \sum_{j=0}^{n+1} (n\varepsilon)^j / j! - (1+\varepsilon)^n$.*

It can be shown that $R$ vanishes in the limit $n \to \infty$.

Consider node $i, j \in V$ and landmark $\lambda \in V$. Let $p_{ij}^\lambda(s)$ denote the probability that the length of paths between $i$ and $j$ via $\lambda$ is at most $s$ for $s = 1, 2, \cdots$. $p_{ij}^\lambda(s)$ is equivalent to the probability that there exists at least one *walk* (i.e., revisiting a node is allowed) of length $s$ from $i$ to $j$ via $\lambda$. The probability of the existence of a walk of length $s$ in a specific node sequence $i \to v_1 \to \ldots \to v_{s-1} \to j$ is given by

$$q_{iv_1}q_{v_1v_2}\cdots q_{v_{s-1}j}$$

where $q_{ij}$ is the edge probability as defined in (3). We claim that $p_{ij}^\lambda(s)$ can be expressed as (11). The first summation in the bracket of (11) counts all possible walks of length $s$ from $i$ to $j$ and sums up their probabilities. The second summation in the bracket of (11) counts all possible walks from $i$ to $j$ but never visits $\lambda$. Thus the subtraction in the bracket counts all the walks from $i$ to $j$ visiting $\lambda$ at least once. Thus, expression (11) is the probability of the existence of walks of length $s$ from $i$ to $j$ via $\lambda$ from Lemma 1, e.g., $A_k$ in Lemma 1 corresponds to an event of a walk. The expression is asymptotically accurate in $N$, i.e., although Lemma 1 requires events $A_k$ be independent, and the same edge may participate between different $A_k$'s and induce correlation, the fraction of such correlations becomes negligible when $s \ll N$, as argued in [17].

Let us evaluate the summations in the bracket of (11). We have

$$\sum_{v_1=1}^N \cdots \sum_{v_{s-1}=1}^N q_{iv_1}q_{v_1v_2}\cdots q_{v_{s-1}j} = h_i h_j \frac{\left(N\langle h^2\rangle\right)^{s-1}}{\beta^s}$$

whereas

$$\sum_{\substack{v_1=1 \\ v_1\neq\lambda}}^N \cdots \sum_{\substack{v_{s-1}=1 \\ v_{s-1}\neq\lambda}}^N q_{iv_1}q_{v_1v_2}\cdots q_{v_{s-1}j} = h_i h_j \frac{\left(N\langle h^2\rangle - h_\lambda^2\right)^{s-1}}{\beta^s}$$

Thus the subtraction in the bracket of (11) is given by

$$\frac{h_i h_j}{\beta^s}\left[\left(N\langle h^2\rangle\right)^{s-1} - \left(N\langle h^2\rangle - h_\lambda^2\right)^{s-1}\right]$$

$$= \frac{h_i h_j}{\beta^s}\left(N\langle h^2\rangle\right)^{s-1}\left[1 - \left(1 - \frac{h_\lambda^2}{N\langle h^2\rangle}\right)^{s-1}\right] \quad (12)$$

$$\approx \frac{h_i h_j}{\beta^s}\left(N\langle h^2\rangle\right)^{s-1}(s-1)\frac{h_\lambda^2}{N\langle h^2\rangle}$$

$$= \frac{h_i h_j h_\lambda^2}{\beta N\langle h^2\rangle}(s-1)\left(\frac{N\langle h^2\rangle}{\beta}\right)^{s-1} \quad (13)$$

Let random variable $L_{ij}(\lambda)$ denote the path length from node $i$ to $j$ with visiting landmark $\lambda$. Then we have $p_{ij}^\lambda(s) = P(L_{ij}(\lambda) \leq s)$. Let

$$F_\lambda(s) := P(L_{ij}(\lambda) > s).$$

We have that

$$F_\lambda(s) = 1 - p_{ij}^\lambda(s) = \exp\left[-\frac{h_i h_j h_\lambda^2}{\beta N\langle h^2\rangle}(s-1)\left(\frac{\langle h^2\rangle N}{\beta}\right)^{s-1}\right]$$

for $s = 1, 2, \cdots$.

Consider the minimum distance among the routes via landmarks $\lambda_k$, $k = 1, \cdots, K(N)$ where the landmarks are chosen i.i.d. from distribution $\sim Q$. Let $L_{ij}$ denote the minimum distance among the routes from $i$ to $j$ via the landmarks. Then

$$L_{ij} = \min[L_{ij}(\lambda_1), L_{ij}(\lambda_2), \cdots, L_{ij}(\lambda_{K(N)})].$$

We have that

$$P(L_{ij} > s)$$
$$= P(\min[L_{ij}(\lambda_1), L_{ij}(\lambda_2), \cdots, L_{ij}(\lambda_{K(N)})] > s)$$
$$= P(L_{ij}(\lambda_1) > s, L_{ij}(\lambda_2) > s, \cdots, L_{ij}(\lambda_{K(N)}) > s)$$
$$= \prod_{k=1}^{K(N)} P(L_{ij}(\lambda_k) > s)$$
$$= \exp\left[-\frac{h_i h_j}{\beta N\langle h^2\rangle}\left(\sum_{k=1}^{K(N)} h_{\lambda_k}^2\right)(s-1)\left(\frac{\langle h^2\rangle N}{\beta}\right)^{s-1}\right]$$
$$= \exp\left[-\frac{h_i h_j}{\beta N\langle h^2\rangle}K(N)\cdot\langle h^2\rangle_Q\cdot(s-1)\left(\frac{\langle h^2\rangle N}{\beta}\right)^{s-1}\right]$$

which proves (1), where we have used

$$\left(\sum_{k=1}^{K(N)} h_{\lambda_k}^2\right) = K(N)\cdot\frac{1}{K(N)}\left(\sum_{k=1}^{K(N)} h_{\lambda_k}^2\right)$$
$$= K(N)\cdot\langle h^2\rangle_Q$$

in the last equation, and $\langle\cdot\rangle_Q$ denotes the expectation of the hidden variables of landmarks chosen according to $\sim Q$ assuming $K(N)$ is sufficiently large.

$$-\frac{\mathrm{Ei}\left(-\frac{a}{\log b}\right)}{\log b} = \frac{-\gamma - \log a + \log\log b}{\log b} \tag{14}$$

$$= \frac{-\log(h_i h_j) - \log\left(\langle h^2 \rangle_Q K(N)\right) + \log(N\beta\langle h^2 \rangle) + \log\log\left(\frac{N\langle h^2 \rangle}{\beta}\right) - \gamma}{\log N + \log\langle h^2 \rangle - \log\beta} \tag{15}$$

## A.2 Proof of Theorem 2.

Let $l_{ij}$ denote the mean length of the shortest detour via landmarks from $i$ to $j$. We have that

$$l_{ij} = \sum_{s=1}^{\infty} P(L_{ij} > s)$$

$$= \sum_{s=0}^{\infty} \exp\left[-\frac{h_i h_j}{\beta N \langle h^2 \rangle} \cdot \langle h^2 \rangle_Q K(N) \cdot s\left(\frac{\langle h^2 \rangle N}{\beta}\right)^s\right]$$

We utilize the Poisson summation formula [22]:

$$l_{ij} = \frac{1}{2}f(0) + \int_0^{\infty} f(t)\,dt + 2\sum_{n=1}^{\infty}\int_0^{\infty} f(t)\cos(2\pi nt)\,dt \tag{16}$$

where

$$f(t) = \exp\left[-atb^t\right], \tag{17}$$

$$a := \langle h^2 \rangle_Q K(N) \cdot \frac{h_i h_j}{\beta N \langle h^2 \rangle}, \tag{18}$$

$$b := \frac{\langle h^2 \rangle N}{\beta} \tag{19}$$

Firstly we have $f(0) = 1$. Next, we evaluate the second term of (16):

$$\int_0^{\infty} \exp\left(-atb^t\right) dt = \int_0^{\infty} \exp\left(-ate^{t\log b}\right) dt$$

$$= (\log b)^{-1}\int_0^{\infty} \exp\left(-\frac{a}{\log b}te^t\right) dt \tag{20}$$

Let $u = te^t$, then we have

$$dt = \frac{du}{u + e^{W(u)}}$$

where $W(\cdot)$ is the Lambert $W$ function which is the inverse of function $te^t$ for $t \geq 0$. Thus (20) is equal to

$$(\log b)^{-1}\int_0^{\infty} \frac{\exp\left(-\frac{a}{\log b}u\right)}{u + e^{W(u)}}\,du$$

Since $W(u) \geq 0$ for $u \geq 0$, (20) is bounded above by

$$(\log b)^{-1}\int_0^{\infty} \frac{\exp\left(-\frac{a}{\log b}u\right)}{u + 1}\,du$$

$$= (\log b)^{-1}\exp\left(\frac{a}{\log b}\right)\int_1^{\infty} \frac{\exp\left(-\frac{a}{\log b}u\right)}{u}\,du$$

$$= -\exp\left(\frac{a}{\log b}\right)\frac{\mathrm{Ei}\left(-\frac{a}{\log b}\right)}{\log b} \tag{21}$$

where $\mathrm{Ei}(\cdot)$ denotes the exponential integral. Consider the assumption $K(N) = o(N)$, i.e., the number of landmarks is not too large compared to $N$. Under this assumption, one can verify that $a/\log b$

is at most $o(N)/N$ which tends to 0 as $N \to \infty$. Thus the exponential term of (21) can be approximated to 1. Thus, (21) reduces to (15). In (14), we used

$$-\mathrm{Ei}\left(-\frac{a}{\log b}\right) \approx -\gamma - \log\left(\frac{a}{\log b}\right)$$

where the error term associated with exponential integral vanishes because $a/\log b$ is small, and $\gamma \approx 0.5772$ is the Euler's constant. Finally, one can show that the last term of (16) vanishes, by using generalized mean value theorem [17]. By averaging (15) over all $i, j \in V$, we obtain (2) from (16).

## A.3 Proof of Theorem 3

Let $M(N) := (\log N) \cdot K(N)$. The landmarks are selected at random from $M(N)$ nodes with highest values of $h$. This implies that the distribution $Q(\cdot)$ of $h$ values of landmarks is given by the following conditional distribution:

$$Q(t) = \rho\left(t \mid h \geq \frac{1}{\sqrt{M(N)}}\right)$$

$$= \rho(t) \cdot \mathbf{1}\left(t \in \left[\frac{1}{\sqrt{M(N)}}, 1\right]\right)\bigg/P\left(h \in \left[\frac{1}{\sqrt{M(N)}}, 1\right]\right) \tag{22}$$

where $\rho(\cdot)$ is the distribution of $h$ given by (8). From (22), we have that

$$\langle h^2 \rangle_Q = \frac{\left\langle h^2\mathbf{1}\left(h \in \left[\frac{1}{\sqrt{M(N)}}, 1\right]\right)\right\rangle}{P\left(h \in \left[\frac{1}{\sqrt{M(N)}}, 1\right]\right)}$$

We have

$$P\left(h \in \left[\frac{1}{\sqrt{M(N)}}, 1\right]\right) = \int_{\frac{1}{\sqrt{M(N)}}}^{1} \rho(h)\,dh$$

$$= \frac{2}{N}\int_{h=\frac{1}{\sqrt{M(N)}}}^{1} h^{-3}\,dh$$

$$\approx \frac{M(N)}{N}$$

and

$$\left\langle h^2\mathbf{1}\left(h \in \left[\frac{1}{\sqrt{M(N)}}, 1\right]\right)\right\rangle = \int_{h=\frac{1}{\sqrt{M(N)}}}^{1} h^2\rho(h)\,dh$$

$$= \frac{2}{N}\int_{h=\frac{1}{\sqrt{M(N)}}}^{1} h^{-1}\,dh$$

$$\approx \frac{\log M(N)}{N}$$

Thus, we have

$$\langle h^2 \rangle_Q = \frac{\log M(N)}{M(N)}$$

Applying the result to (4), and using $M(N) = (\log N) \cdot K(N)$, we obtain (5).

## B NUMERICAL VERIFICATION OF THEORETICAL RESULTS

### B.1 Detour distances in ER networks

We verify the derived upper bounds on detour distances in ER networks given by (6) using simulation. Fig. 2 shows the comparison between the simulated detour distances and theoretical bounds in (6) in ER networks for varying number of landmarks $K(N)$. We evaluated the cases where $K(N) = \log N$, $N^{0.5}$ and $N^{0.9}$. In all cases, we observe that the derived upper bound provides very good estimates on the actual detour distances.

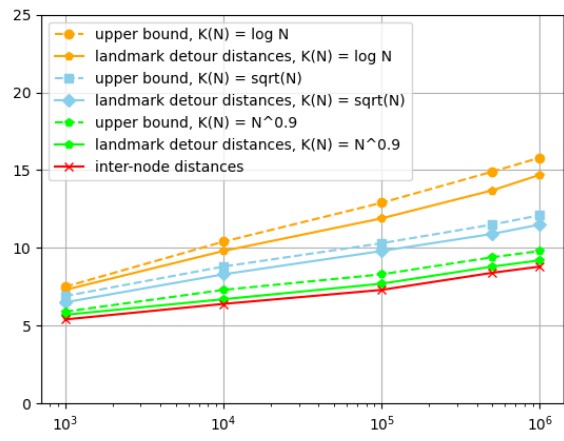

Figure 2: Comparison of inter-node distances, landmark detour distances, and upper bound for $K(N) = \log N, N^{0.5}, N^{0.9}$ in ER networks.

### B.2 Detour distances in BA networks

We verify the derived upper bounds on detour distances in BA networks given by (4). Fig. 3 shows a comparison between the derived upper bounds and the simulated detour distances in BA networks. We observe that the theoretical bound is an excellent match with the simulated distances. In addition, the inter-node distances in BA network are shown in Fig. 3. Indeed, the simulated detour distances and the theoretical bounds are quite close to the shortest path distances, which verifies our theoretical results.

## C ACTUAL TRAINING TIME AND GPU MEMORY USAGE

Table 5 and 6 show the comparison and actual training time and GPU memory usage between vanilla GCN with HPLC. Note that *HPLC already contains vanilla GCN as a component*. Thus, one should attend to the *additional* resource usage incurred by HPLC in Table 5 and 6. Experiments are conducted with NVIDIA A100 with 40GB memory. We observe that, the additional space and

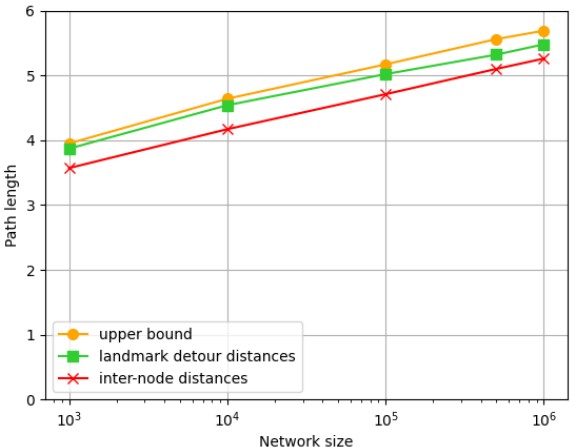

Figure 3: Comparison of inter-node distances, landmark detour distances, and theoretical upper bounds in BA networks.

time complexity incurred by HPLC relative to vanilla GCN is quite reasonable, demonstrating the scalability of our framework.

| Dataset | vanilla GCN | HPLC |
|---------|-------------|------|
| CITATION2 | 282856 (ms) | 352670 (ms) |
| COLLAB | 1719 (ms) | 2736 (ms) |
| DDI | 3246 (ms) | 3312 (ms) |
| PubMed | 8859 (ms) | 8961 (ms) |
| Cora | 982 (ms) | 1022 (ms) |
| Facebook | 1107 (ms) | 1243 (ms) |

Table 5: Comparison of training time for 1 epoch between vanilla GCN and HPLC.

| Dataset | vanilla GCN | HPLC |
|---------|-------------|------|
| CITATION2 | 29562 (MB) | 36344 (MB) |
| COLLAB | 4988 (MB) | 7788 (MB) |
| DDI | 2512 (MB) | 2630 (MB) |
| PubMed | 2166 (MB) | 3036 (MB) |
| Cora | 1458 (MB) | 6160 (MB) |
| Facebook | 2274 (MB) | 2796 (MB) |

Table 6: Comparison of GPU memory usage during training between vanilla GCN and HPLC.

## D NUMBER OF CLUSTERS

Table 7 shows the performance with varying number of clusters. Specifically, $K = \eta \log N$ where we vary hyperparameter $\eta$. The results show that performance can be improved by adjusting the number of clusters.

## E NODE CENTRALITY

For each cluster, the most "central" node should be selected as the landmark. We have used Degree centrality in this paper; however,

| $\eta$ | COLLAB | DDI | PubMed | Cora | Citeseer | Facebook |
|---|---|---|---|---|---|---|
| 1 | 55.42 ± 0.51 | 68.27 ± 6.47 | 96.41 ± 0.13 | 94.34 ± 0.16 | 94.70 ± 0.31 | 99.53 ± 0.00 |
| 2 | 55.44 ± 0.80 | 68.95 ± 7.30 | 96.42 ± 0.17 | 94.51 ± 0.18 | 94.73 ± 0.12 | 99.48 ± 0.00 |
| 3 | 55.76 ± 0.48 | 69.34 ± 7.03 | 96.38 ± 0.22 | 94.61 ± 0.11 | 94.85 ± 0.14 | **99.69 ± 0.00** |
| 4 | 59.95 ± 0.54 | 69.95 ± 8.23 | 96.89 ± 0.19 | 94.68 ± 0.12 | 95.14 ± 0.11 | 99.63 ± 0.00 |
| 5 | **56.04 ± 0.28** | **70.03 ± 7.02** | 97.32 ± 0.17 | 94.74 ± 0.12 | 95.34 ± 0.18 | 99.60 ± 0.00 |
| 6 | - | - | 97.31 ± 0.24 | 94.81 ± 0.19 | 95.87 ± 0.13 | 99.64 ± 0.00 |
| 7 | - | - | **97.38 ± 0.34** | **94.95 ± 0.18** | **96.15 ± 0.19** | 99.65 ± 0.00 |

Table 7: Ablation study with hierarchical graph clustering in terms of hyperparameter $\eta$.

| Centrality | DDI | PubMed | Cora | Citeseer | Facebook |
|---|---|---|---|---|---|
| Degree | **70.03 ± 7.02** | **97.38 ± 0.34** | **94.95 ± 0.18** | **96.15 ± 0.19** | **99.69 ± 0.00** |
| Betweeness | 69.71 ± 6.87 | 97.16 ± 0.19 | 94.69 ± 0.12 | 95.84 ± 0.08 | 99.52 ± 0.00 |
| Closeness | 69.56 ± 5.65 | 96.92 ± 0.20 | 94.43 ± 0.10 | 95.65 ± 0.12 | 99.45 ± 0.00 |

Table 8: Link prediction results from landmark selection with different centrality.

| Dataset | FluidC | Louvain |
|---|---|---|
| COLLAB | **56.04 ± 0.28** | - |
| DDI | **70.03 ± 7.02** | 60.79 ± 8.89 |
| Pubmed | **97.38 ± 0.34** | 95.83 ± 0.23 |
| Cora | **94.95 ± 0.18** | 93.78 ± 0.15 |
| Citeseer | **96.15 ± 0.19** | 94.98 ± 0.12 |
| Facebook | **99.69 ± 0.00** | 99.30 ± 0.00 |

Table 9: Comparison of graph clustering algorithms.

there are other types of centrality such as Betweenness and Closeness. Betweenness centrality is a measure of how often a given node is included in the shortest paths between node pairs. Closeness centrality is the reciprocal of the sum-length of shortest paths to the other nodes.

Table 8 shows the experimental results comparing Degree, Betweenness and Closeness centralities. The results show that the performances are similar among the centralities. Thus, all the centralities are effective measures for identifying "important" nodes. Degree centrality, however, was better than the other choices.

More importantly, Betweenness and Closeness centralities require full information on inter-node distances, which incurs high computational overhead. In Table 8, we excluded datasets CITATION2 and COLLAB which are too large graphs to compute Betweenness and Closeness centralities. Scalability is crucial for link

prediction methods. Thus we conclude that, from the perspective of scalability and performance, Degree centrality is the best choice.

## F GRAPH CLUSTERING ALGORITHM

We conducted ablation study such that FluidC is replaced by Louvain algorithm [7] which is widely used for graph clustering and community detection. Table 9 shows that our model performs better with FluidC than with Louvain algorithm in all datasets. The proposed hierarchical clustering using FluidC can control the number of clusters to achieve a good trade-off between complexity and performance. However, Louvain algorithm automatically sets the number of clusters, and the number varied significantly over datasets. Moreover, for COLLAB dataset, Louvain algorithm could not be used due to resource issues. Thus, we conclude that FluidC is the better choice in our framework. We show the performance with varying number of clusters of FluidC in Appendix D.

## G DATASETS

Dataset statistics is shown in Table 10.

## H HYPERPARAMETERS

The hyperparameters for experiments are shown in Table 11.

| Dataset | # Nodes | # Edges | $\frac{\text{\#Edges}}{\text{\#Nodes}}$ | Avg. node deg | Density | Split ratio | Metric |
|---|---|---|---|---|---|---|---|
| Cora | 2,708 | 7,986 | 2.95 | 5.9 | 0.0021% | 70/10/20 | AUC |
| Citeseer | 3,327 | 7,879 | 2.36 | 4.7 | 0.0014% | 70/10/20 | AUC |
| PubMed | 19,717 | 64,041 | 3.25 | 6.5 | 0.00033% | 70/10/20 | AUC |
| Facebook | 4,039 | 88,234 | 21.85 | 43.7 | 0.0108% | 70/10/20 | AUC |
| OGB-DDI | 4,267 | 1,334,889 | 312.84 | 500.5 | 14.67% | 80/10/10 | Hits@20 |
| OGB-CITATION2 | 2,927,963 | 30,561,187 | 10.81 | 20.7 | 0.00036% | 98/1/1 | MRR |
| OGB-COLLAB | 235,868 | 1,285,465 | 5.41 | 8.2 | 0.0046% | 92/4/4 | Hits@50 |

**Table 10: Dataset statistics.**

| Hyperparameter | Value |
|---|---|
| Encoder of all plug-in methods | GCN |
| Learning rate | 0.001, 0.0005 |
| Hidden dimension | 256 |
| Number of GNN layers | 2, 3 |
| Number of Decoder layers | 2, 3 |
| Negative sampling | Uniformly Random sampling |
| Dropout | 0.2, 0.5 |
| Negative sample rate | 1 |
| Activation function | ReLU (GNNs), LeakyReLU ($f_{i(k)}$) |
| Loss function | BCE Loss |
| Use edge weights | False (only binary edge weights) |
| The number of landmarks | $O(\log N)$ |
| Optimizer | Adam [25] |

**Table 11: Detailed hyperparameters.**

