# OpenReview forum: "Hierarchical Position Embedding of Graphs with Landmarks and Clustering for Link Prediction"
_ACM.org/TheWebConf/2024/Conference — TheWebConf24_

### Official Review · Reviewer_YFYA · 2023-10-31

**Novelty:** 3
**Technical Quality:** 3

**Review:**

This paper proposes a node embedding method for link prediction task, which takes local potional information into consideration by calculating the distance between nodes and local landmarks after clustering. The authors provide sufficient theoretical foundation of how many and what kind of landmarks should be chosen in different kind of graphs. The authors compare the proposed method against fifteen related methods on seven real-world datasets demonstrating a promising result. Overall, the paper is well organized and easy to follow.

Pros:
(1) This paper is easy to follow. The method description, and the experiments are clear in logic.
(2) The results show the promising of the proposed approach

Cons:
(1) This paper is less motivated. In the section of Introduction, the authors divide current position-based methods into two categories and summarize the limitations as "fall short of state-of-the-art", "does not scal well" and "may not outperform" without clear elplanation. Besides, I really want to know why the proposed method overcomes the current dilemma but cannot find such description.
(2) The experiments are insufficient. There are only experiments to validate the effectiveness of HPLC. The authors claim that HPLC is efficient (at line 94) and robustness (at line 151). However, no experimental results can support this conclusion. Besides, the ablation study is also insufficient. I want to know how does HPLC performs with only CE or MV.
(3) Theoritical analysis in section 2 seems less related to the method. If I understand correctly, the authors analyze landmarks in three kind of graphs with different distribution of node degree and propose that the core question is how many and what kind of landmarks should be chosen. However, after analysis, I still do not know why should choose high-degree nodes as landmarks.
(4) Landmark graph is designed to be a complete graph, where nodes are fully-connected with each other. I think membership encoding based on such a full connection may affect the distinguishability between nodes.

**Questions:**

(1) What are the limitations of current position-based competitors and why HPLC is better than them?
(2) There are two important hyperparameters: K and R, which are resulted from wo simple formulas. Why do such operations?

**Reviewer Confidence:**

3: The reviewer is confident but not certain that the evaluation is correct

**Scope:**

4: The work is relevant to the Web and to the track, and is of broad interest to the community

---

### Official Review · Reviewer_12xJ · 2023-11-17

**Novelty:** 4
**Technical Quality:** 4

**Review:**

This research presents a method for representing positional information through the use of representative nodes known as landmarks. The researchers delve into combining landmark selection with graph clustering, leveraging on the positional data of nodes in order to boost performance levels in link prediction. They select landmarks and organize the graph in a principled way, unlike previous methods using random selection.

**Strengths:**

1. The authors employ an approach to selecting landmarks, specifically choosing nodes with high degree instead of random selection. Furthermore, they provide a theoretical justification for this method.

2. Extensive experimental analysis on 8 datasets demonstrates positive outcomes, indicating the potential applicability of the proposed method in link prediction tasks.

**Weaknesses:**

1. Reproducibility issue. The source code has not been shared.

2. The implementation details of the position encoder are not well described. Please provide more details of the position encoder.

3. The experimental analysis of the ablation experiment is less clear. Please provide relevant explanations and details of the ablation experiment.

**Questions:**

See the weaknesses part.

**Reviewer Confidence:**

3: The reviewer is confident but not certain that the evaluation is correct

**Scope:**

4: The work is relevant to the Web and to the track, and is of broad interest to the community

---

### Official Review · Reviewer_ZZpU · 2023-11-23

**Novelty:** 5
**Technical Quality:** 5

**Review:**

Summary
This paper investigates the hierarchical position encoding of graphs with landmarks.  Position encoding encodes a node by exploiting the distances to a set of select nodes named landmarks. The authors provide a detailed theoretical analysis of the strategy of selecting landmarks on various types of random graphs. With the insights, the authors then propose a hierarchical position encoding, in which they first partition the graph into small clusters and select landmark nodes in each cluster separately. Experiments show that the proposed method achieves sota performance.

Strong points
The paper is well-written and easy to follow.
The theoretical analysis is solid. The insights may be useful for other researchers.


Opportunities for improvement
The whole analysis focuses on how the path length via the landmarks resembles the actual distance. But it is not obvious whether a tight lower bound/upper bound leads to a good position encoding. For instance, if all nodes are selected as landmarks, the path length via landmarks would be the actual distance. It would be interesting to see if the authors can test this  on a toy graph.

**Questions:**

The whole analysis focuses on how the path length via the landmarks resembles the actual distance. But it is not obvious whether a tight lower bound/upper bound leads to a good position encoding. The quality of position encoding perhaps could be more related to how the landmarks distribute in the graph.
For instance, if all nodes are selected as landmarks, the path length via landmarks would be the actual distance. It would be interesting to see if the authors can test this  on a toy graph.

**Reviewer Confidence:**

2: The reviewer is willing to defend the evaluation, but it is likely that the reviewer did not understand parts of the paper

**Scope:**

4: The work is relevant to the Web and to the track, and is of broad interest to the community

---

### Official Review · Reviewer_9ma2 · 2023-11-27

**Novelty:** 3
**Technical Quality:** 5

**Review:**

Message passing GNNs are known to be as powerful as 1-WL. Hence, there are cases where they can not distinguish between non-isomorphic graphs. In this work, each node is enhanced with distance information with respect to a set of landmark nodes that are chosen as high-degree nodes. The idea of enhancing node information with position/or distance information is not new, however the approach of using landmark nodes and an embedding of landmarks as cluster centroids is novel. The proofs are modified version of [17], adapted to the landmark setting.

Pros:
* The paper is in general well written and easy to follow.
* The results regarding E-R and B-A are not surprising, but act as a sanity check and inspiration for the applicability of this method.
* The results on link prediction outperform a wide variety of traditional link prediction baselines, as well more recent GNN based approaches.
* There is substantial experimental verification of most claims in this work (theoretical, ablation, etc.).

Cons:
* The exact setting for link prediction is not clearly described. While a subset of edges is chosen for training, the amount of negative samples (non-edges) is not described. More precisely, how do the authors choose the negative samples, is it the whole set of  non-edges (in this case the d/s is highly imbalanced) or a subset of them? Moreover, different measures are used to evaluate the methods for each dataset, making the comparison rather confusing.
* There is prior work on landmark selection that is not discussed, some with theoretical results -- mostly related to the NP-completeness on choosing landmarks for general graphs. (E.g., Potamias et al., "Fast shortest path distance estimation in large networks" &  Zhao et al., "Orion: Shortest Path Estimation for Large Social Graphs").
* The evaluation is only performed for link prediction, while it can be easily applied to node classification.
* The proof in A1 relies on a lemma that is applied to mutually independent events, however the events under consideration are not independent. There is a discussion on that "the fraction of such correlations becomes negligible", however this is more an intuitive explanation rather a concrete setting. Also, the Bernoulli approximation is applied in Eq.(12). For it to be correct there should be conditions on $\frac{h_\lambda^2}{N<h^2>} \cdot (s-1)$ that should be stated.

**Questions:**

Q1: How is the set of non-edges chosen? Is it ${N \choose 2} - E$, or a random sample of them?

Q2: How are clusters merged to macro-clusters in 3.3.?

**Reviewer Confidence:**

3: The reviewer is confident but not certain that the evaluation is correct

**Scope:**

4: The work is relevant to the Web and to the track, and is of broad interest to the community

---

### Official Review · Reviewer_eTou · 2023-12-01

**Novelty:** 4
**Technical Quality:** 5

**Review:**

The authors consider the task of link prediction using graph neural networks (GNNs). It has recently been proposed to use positional information in the node embedding in GNNs. The authors propose a scalable way to do so using a cluster + landmark strategy. They also provide an analysis on landmark selection in random graph models that is complementary to the positional embedding contribution. They demonstrate strong empirical results for link prediction tasks.

*Note:* I have reviewed a previous version of this paper for a different venue and used my previous review as a starting point for this review. I have updated the relevant portions of the review to account for changes made by the authors since the previous submission.

*After author rebuttal:* My opinion on this paper hasn't changed much. There's nothing particularly wrong with it. As the authors point out, ad-hoc innovations leading to better performance are indeed a valuable contribution. There's also nothing that particularly excites me about it, but it is solid research that I would hope to see published in a reputable venue.

## Strengths
- Interesting analysis on landmark selection in both the Erdős-Rényi and Barabási-Albert (B-A) models that provides insights on how to choose landmarks on such Poisson and scale-free networks, respectively. These results could be of independent interest to the network science community.
- Many small innovations leading to strong empirical performance on a variety of data sets compared to lots of other methods. Ablation studies are also provided to justify the need for the different innovations.

## Weaknesses
- While the analysis in Section 2 is rigorous and principled, a lot of the proposed innovations in Section 3 are very much ad-hoc with lots of choices of hyperparameters. While I still consider this a weakness, this area has improved from the previous submission, where the authors now try to connect some of the design decisions in Section 3 with the analysis in Section 2, e.g., in the paragraph labeled "Effect of Clustering on Landmarks".

### Minor presentation issues:
- Equations (1) and (2) come after equation (3) in the paper.
- Possible error in bolding in Table 7: $\eta = 4$ with COLLAB looks like it has higher accuracy than for $\eta = 5$.

**Questions:**

1. In most of the 6 data sets in Table 7, the accuracy looks to increase with $\eta$. Does this trend continue in general? Is there a reason to stop increasing $\eta$ aside from increasing computation time and memory requirement?

**Ethics Review Description:**

No concerns

**Reviewer Confidence:**

3: The reviewer is confident but not certain that the evaluation is correct

**Scope:**

3: The work is somewhat relevant to the Web and to the track, and is of narrow interest to a sub-community

---

### Decision · Program_Chairs · 2024-01-22

**Decision:**

Accept

**Comment:**

The paper makes a solid contribution in the positional embedding of graphs using landmarks. The authors have responded to reviewer comments in an exhaustive manner.